# Modeling Turkish Households' Climate Change-Related Behaviors: Theory of Planned Behavior Approach

**Ahmad Samim Pouya and Özge Can Niyaz \*** 

Department of Agricultural Economics, Faculty of Agriculture, Canakkale Onsekiz Mart University, Canakkale 17100, Turkey
\* Correspondence: ozgecanniyaz@comu.edu.tr; Tel.: +90-286-218-0018; Fax: +90-286-218-0545

**Abstract:** Global environmental problems are both the cause and outcome of human actions. Even though families contribute significantly to the problem, little is known about the reasons for household climate change behavior. Prior research has shown that household intentions and behavior play an essential role in climate change adaptation and mitigation. The goal of this exploratory study was to see how climate change-related factors influenced climate change-related intentions and behaviors. In terms of climate change, Turkey is a vulnerable country in Mediterranean Europe. As a result, the goal of this study is to apply the Theory of Planned Behavior to simulate Turkish households' climate change-related behavior. Using a random sampling method, an online self-reported questionnaire of 400 Turkish households assessed the impact of practices to adapt and mitigate climate change. Within the context of the Theory of Planned Behavior, Structural Equation Modeling was used to examine household attitudes and behaviors about climate change. The findings imply that household intentions are important predictors of climate change-related behavior in Turkey. In addition, subjective norms and perceived behavioral control influence the goals of Turkish households. As a result, efforts should be undertaken to provide households with the subjective and perceptual abilities and tools they need to manage their climate-related activities.

**Keywords:** behavior; climate change; structural equation method; theory of planned behavior

## 1. Introduction

Environmental problems can be classifiable as water, air, and soil pollution; improper waste disposal; deforestation; overpopulation; increasing resource usage; and global warming [1–5]. Climate change is mainly caused by global warming [4,6–9] and environmental problems are the underlying reasons [10,11]. Climate change has been an important issue since the 1880s, beginning from the Industrial Revolution to today [12,13]. Every decade since the 1880s, the earth's temperature has increased by 0.14 °F (0.08 °C) and especially in the past 40 years, this increase has been rising to values more than double the increase mentioned above at 0.32 °F (0.18 °C) per decade [13]. On a global scale, 2020 was the second-warmest year in recorded history [13,14]. Therefore, climate change is an increasingly important problem in the world these days [15–19]. Evidence suggests that climate change is among the most important factors influencing global economies [2,16,20–23]. In addition to its effect on economies, climate change has a significant role in the interaction of environmental problems, human populations, and agriculture as well [24–30].

The world population is growing daily [31,32] and if preventive measures are not taken, the climate change problem will affect an ever-increasing amount of people [30,33]. Shortly, probable outcomes related to climate change include economic losses, the destruction of natural resources, a decline in bio-diversity, increases in mortality, famine, and widespread scarcity [34–36].

Environmental concerns related to environmental issues increase with each passing day [5,16,37–39]. Climate anomalies, such as temperature extremes, drought, wind-related weather phenomena, wildfires, and flooding are more and more noticeable in

recent years [40–42]. Global warming is highly correlated to human activity, which creates greenhouse gas emissions from fossil fuels, driving habits, farming, overconsumption, and extreme production in the main sectors (industry, agriculture, service), etc. [2,4,15,20,23,43–47]. Previous literature shows that it is important to address individuals' behavior to obtain solutions for adaptation and the mitigation of climate change. However, the number of studies addressing household behavior related to climate change is very limited.

Climate change effects could be reduced by applying action plans that contain implementations, such as incentives for less consumption, reuse of materials, recycling, and afforestation. Developed economies and associations, such as the European Commission (EC), United Nations (UN), United States Environmental Protection Agency (USEPA), and the World Bank (WB) already have action plans whose common aims are to advance the climate change aspects of greener economies [48–51].

The Paris Agreement of 2015 is a treaty with the exclusive aim of adapting to global climate change as per the context of the United Nations Framework Convention on Climate Change. Turkey was the 192nd country that signed the Paris Agreement for adaptation to climate change at the end of 2021. It aims to reduce human-induced greenhouse gas emissions in line with the agreement. Turkey expects a reduction in its greenhouse gas emissions by up to 21% by 2030 within the framework of the Paris Agreement [40]. As a result, research in this field is becoming increasingly important to meet the aims of global and national action plans. However, there are no studies modeling household climate change behaviors in Turkey that will serve to achieve the Paris Agreements' 2030 goals. Therefore, the main question of this study is, "What are the drivers affecting the intention and behavior of Turkish households regarding climate change?". In this context, this study aims to analyze Turkish households' climate change-related behaviors with the help of the Theory of Planned Behavior (TPB).

## 2. Literature Review, Theoretical Background and Hypothesis

Carbon emissions are closely related to private household consumption and individual choices and behaviors [16,52]. Therefore, incentivizing individuals to adopt more sustainable behavior is an important policy objective [2]. Although climate change may be largely anthropogenic, many environmental actions aimed at adaptation and/or the mitigation of climate change can directly impact the global climate change cycle by limiting human impact on the local environment [4,20].

In addition, the benefits of behavioral change may also be considered insignificant and might only occur when many people change their behavior. Global warming illustrates these difficulties [18,38,45]. Previous literature shows that climate change is mainly a human-made and preventable issue. Therefore, studies on human behavior are even more important than in the past [1,5,9,15,17,18,21–23,44,46,47].

Household behavior is a complex, multidimensional phenomenon that is compounded by the addition of environmental concerns [2]. The Theory of Planned Behavior is a widely used model to explain households' environmental behavior [53–62]. TPB has been used to examine several environmental issues, including littering [63–65], recycling [56,66,67], energy conservation, and carbon reduction [68], and general environmental behavior [69–71].

However, few studies have addressed climate change-related behaviors as direct effects on attitudes, subjective norms of climate change behaviors, or perceptions of behavioral control [8,38,39,68,72–74]. It is important to highlight that studies have yet to be conducted with this method in Turkey.

TPB (Figure 1) is a popular theory of behavior change used to better understand human behavior and its antecedents. TPB emphasizes the role of attitudes, subjective norms, and perceived behavioral control in shaping behavioral intentions, which together explain the considerable variation in individuals' actual behavior [75,76]. Despite its broad appeal, scholars continue to refine the model and test its applicability across topics and contexts [77]. However, there are still notable gaps in the theory. For example, existing

academic research points to the need for more understanding of the impact on key components of the theory [78]. To date, there has been insufficient research on the origin or impact of attitudes toward climate change-related behavior, Subjective Norms (SN), and Perceived Behavioral Control (PBC).

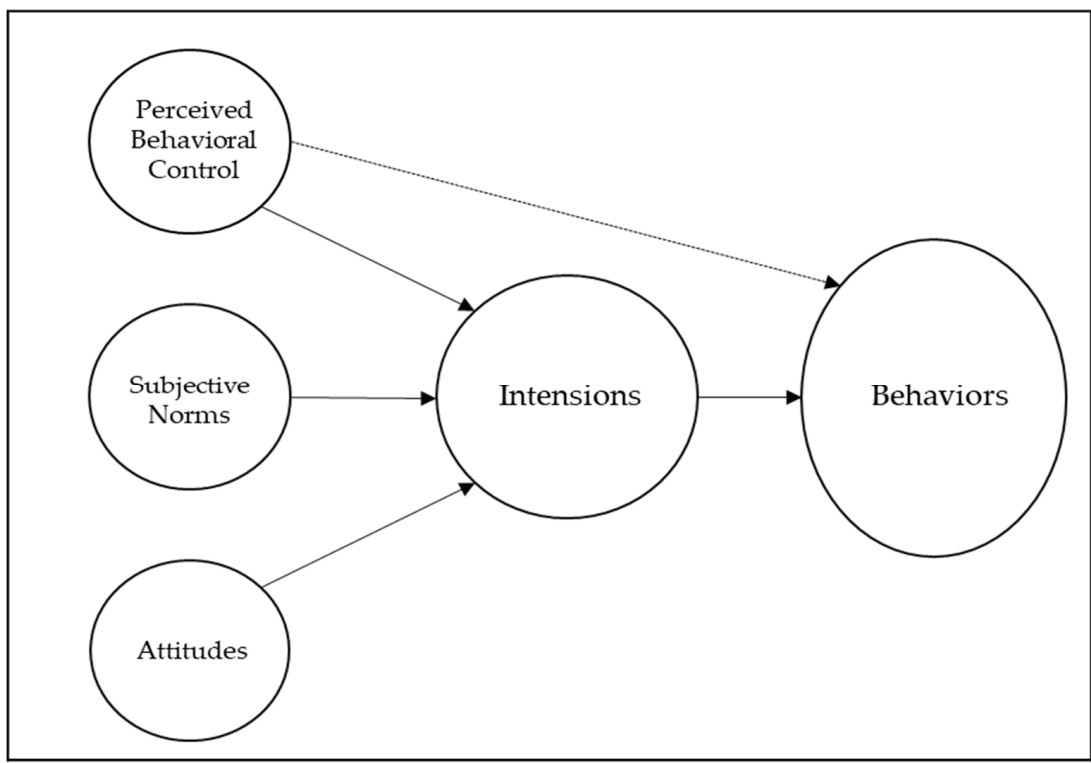

**Figure 1.** Ajzen's Theory of Planned Behavior. References: [75,76].

Impressive communication with Turkish families leads to criticism of adapting to changing conditions; however, this can be a very problematic issue unless the attitudes towards climate change-related behavior can be comprehended properly [8,15]. The predictors we propose come from Ajzen's Theory of Planned Behavior. In this context, the climate change behavior of Turkish households was modeled with the help of simultaneous equation systems, whether SN, PBC and attitudes were determined within the framework of Planned Behavior Theory in this study. Within the scope of the research, the formulated hypotheses (H1, H2, H3, H4, H5) will be tested with the help of the model.

Intentions, attitude towards the behavior, PBC and SN are effective in the formation of behavior. The first hypothesis (H1) is about the expectation of an intention return to behavior according to Ajzen's TPB model. We expect that the households' intention has a positive impact on their climate change-related behavior parallel to relevant literature.

**H1.** *The intention of the household has an impact on their climate change-related behavior [8,72,73].*

Afterward, the main variables that are expected to affect intention, according to Ajzen's theory, are formulated. The probable drivers of the intentions are attitudes, SN and PBC. Attitudes are the tendencies of the individual to exhibit the behavior. Attitudes are expected to be significant on intention according to Ajzen's TPB (H2).

**H2.** *Attitudes towards climate change have an impact on households' intentions [8,72,73].*

SN refers to the social perception of whether or not the behavior is performed and is a social factor that indicates oppression. SN includes beliefs about what others will think of his or her behavior, and the person's extent to which it conforms to expectations influences its intention. It has been formulated whether the opinions of others about climate change have an effect on Turkish household intention (H3) in this research.

**H3.** *Subjective norms of climate change-related behavior have an impact on households' intentions [8,72,73].*

PBC is a variable that indicates whether or not individuals own control of performing any behavior. PBC is a different variable compared to SN and attitudes that can affect not only intention but also behavior. In this context, in this study, the PBC variable is expected to be effective on both intention (H4) and behavior related to climate change (H5).

**H4.** *PBC has an impact on households' intentions [8,72,73].*

**H5.** *PBC has an impact on households' climate change-related behaviors [8,72,73].*

### 3. Material and Methods

*3.1. Research Design, Questionnaire, Variables, and Scale*

In this study, a quantitative approach was used to examine the TPB-related climate change behavior of Turkish households. A structured online questionnaire was created using Google Forms. Google Forms applied a structured online questionnaire between February and July 2020. The online questionnaire was developed based on previous research on climate change-related behavior and TPB applications [8,68,73,79–82].

Data collected from structured questionnaires included information on household socio-demographic characteristics and contextual variables related to household climate change. The details of the variables; variable name, variable group, and frequency of answers associated with the variable group are given in the results section In addition, the questionnaire also included items with potential variables of TPB (climate change-related behaviors, intentions, attitudes, subjective norms, PBC) as pre-literature [8,68,73,79–82].The Likert Scale [83] is widely used in the social sciences and was used to measure the same TPB-related items as in previous studies [8,38,39,68,72–74]. 5-Point Likert Scale (5 = strongly agree . . . 1 = strongly disagree) is used to show item's mean score in this study.

*3.2. Research Area and Sampling*

Turkey is quite vulnerable to climate change. Turkey is located in the southern regions of Mediterranean Europe. Turkey is already affected by climate change, facing high temperatures, floods, fires, and rapid-changeable weather conditions. This has a significant negative impact on the availability of water for food production as well as rural developments, which creates additional social and regional along with national disparities [84,85].

Turkey ranks 17th in the world according to population, with nearly 84 Million in population in 2020. Turkey is also a developing economy in the world. The country was the 21st term of the world economy in 2021 [86,87]. The country is also an important producer and exporter of agricultural products in the global market, and is estimated to be the 7th largest agricultural producer in the world, and the largest producer and exporter of crops such as hazelnuts, chestnuts, apricots, cherries, figs, olives, quinces, tobacco, and tea [88]. Due to the grand population and economic size, Turkey was in the top 20 countries of carbon emission producers in 2020 [89]. Figure 2 shows the place of Turkey (marked with star) in the world and the status of its $CO_2$ emissions at the same time in 2020 [90]. Turkey is in the 6th class (between 2–5 t) according to $CO_2$ emissions from the burning of fossil fuels for energy and cement production (Figure 2) [90].

Random probability sampling was used to obtain an adequate population sample in Turkey [4,91]. According to the Turkish Statistical Institute, the population of Turkey (2020) was 83,614,000 people [92]. With a sampling error of 5%, a confidence interval of 95%, and a standard population proportion of 50%, the minimum sample size was 383.

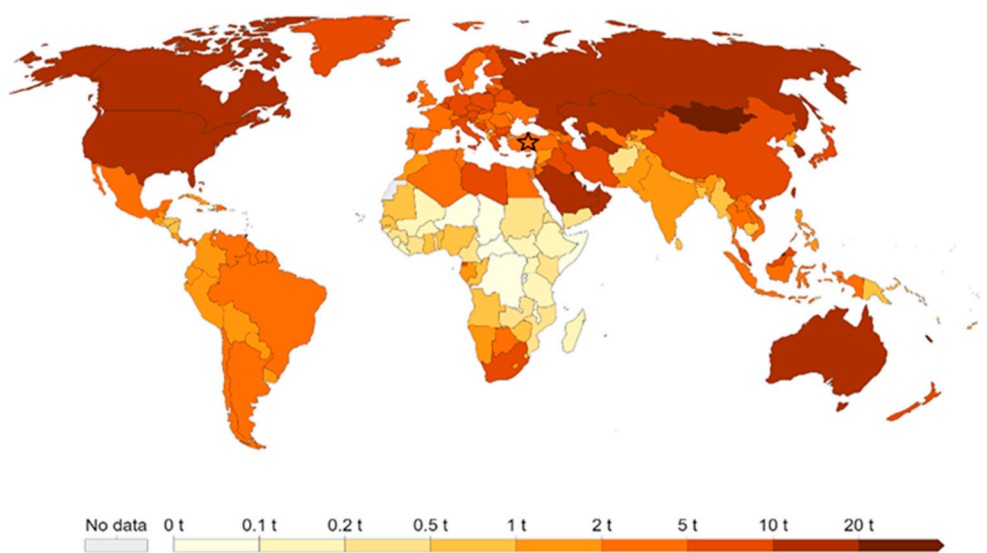

**Figure 2.** CO$_2$ Emissions per Capita Classifications of Turkey in 2020. Reference: [90].

A total of 405 Turkish households participated in the survey. The characteristics of five questionnaires were unreliable or suboptimal. Thus, 400 questionnaires were found to be reliable and valid at the end of the questionnaire phase. Participants in the survey mostly (96.2%) came from the three most populous cities in Turkey, namely Istanbul (58.3%), Ankara (21.3%), and Izmir (16.6%). The three biggest provinces, Istanbul, Ankara, and Izmir, accounted for 30.5% of the Turkish population during the relevant period. There are also valid questionnaires in other cities in Turkey (3.8% of the total questionnaires).

Table 1 provides a summary of the sample demographic data compared to Turkey's general population in 2020. Variables such as age, gender distribution, income distribution, and education level of the basic population related to the Turkish population were taken from the official statistics website [92]. Therefore, when examining the population values, it was concluded that the sample drawn represented Turkey very accurately, except for age and education level. For this reason, the study area is set in Turkey. During the research period, the age range was specifically narrowed to those between 18 and 66 years old. People with higher education participated in the online survey. The level of households' education in online surveys can be higher than in face-to-face survey types [93]. Therefore, the educational level of the respondents is much higher than the Turkish average.

**Table 1.** The characteristics of the respondents in the general population.

| Characteristics | Population | Sample |
| --- | --- | --- |
| Age | Under 18 years (27.2%)<br>Between 18 and 64 years (63.3%)<br>65 years and up (9.5%) | Between 18 and 66 years (100%) |
| Household size (mean) | 3.3 person | 3.0 person |
| Income of households (mean) | Annual mean = 9907 $ *<br>Monthly mean = 825 $ | 714 $ and less = 34.3%<br>715–1.428 $ = 36.7%<br>1.429+ $ = 18.5%<br>No reply = 10.5% |
| Gender | 49.9% of women,<br>50.1% of men | 47.3% of women,<br>52.7% of men |
| Area of residence | 93.0% of city and province centers | 97.7% of city and province centers |
| Education | Low (8 or fewer years) = 44.7%<br>Middle (9–12 years) = 39.7%<br>High (13 and + years) = 15.6% | Low (8 or fewer years) = 14.0%<br>Middle (9–12 years) = 37.8%<br>High (13 and + years) = 48.2% |

* 1 $ = Average 7 Turkish Liras in 2020 [94]. Reference: Population data derived from the [92].

*3.3. Analysis Methods*

The evaluation of the primary data obtained from the questionnaires was started after the completion of the research questionnaires. Statistical Package for Social Science (SPPS) was used in the evaluation of the statistical summaries (means, frequency tables, chi-square, etc.) and Reliability Analysis. In addition, LISREL package programs were used for Confirmatory Factor Analysis and Structural Equation Analysis.

The Reliability Analysis was used to test the reliability and validity of the 5-Point Likert Scale [83] items. The Reliability Analysis is an analysis method used to measure the consistency between the answers to test how reliable and valid the answers that the respondents gave to the survey questions. It can be seen that the higher the correlation calculated as the reliability coefficient of the test, the more consistent and valid the answers given to the questionnaire. The Cronbach Alpha coefficient is widely used in reliability analysis [72,95]. Accordingly, the Cronbach Alpha coefficient:

- $0.00 \leq \alpha < 0.40$, scale is not reliable,
- $0.40 \leq \alpha < 0.60$, scale has low reliability,
- $0.60 \leq \alpha < 0.80$, scale is quite reliable,
- $0.80 \leq \alpha < 1.00$, scale is highly reliable [96,97].

The Structural Equation Model (SEM) is a special and inclusive statistical analysis method that is utilized to analyze structural relationships by testing models where causal and interrelationships between observed and latent variables coexist. SEM situates a fully inclusive model that allows the testing and measuring of meaningful theories by examining the structural relationships. SEM is comprised of a set of statistical methods that conducts a multivariate analysis of the structural theory of the subject being examined by testing a relevant hypothesis. This structural theory illustrates the causal processes observed in many variables. SEM, similar to simple regression analysis, is a multivariate approach that models the interactions between theoretical constructs by incorporating measurement errors into constructs, and relationships between errors [98,99]. SEM consists of two main stages. These are known as the measurement model and the structural model. The measurement model is the model in which the latent variables are estimated with the observed variables. The measurement model shows the relationships between the latent variables and observed variables. The structural model is the model in which the relationships between latent variables are evaluated. The structural model shows the causal relationships between dependent and independent variables. Although the measurement model is used in Confirmatory Factor Analysis (CFA) models, structural models are used in path analysis. CFA is used to test the existence of the theoretical structure. If the model fits the goodness criteria and is suitable as a result of CFA, the next stage, SEM, can be passed. SEM enables causal relationships to be expressed with structural equations [100,101].

The Structural Equation Model, which will be based on the Planned Behavior Theory, will include factors that are expected to have an indirect or direct effect on both Turkish households' climate change-related intentions and behaviors. First of all, within the framework of Planned Behavior Theory, the intention factor is calculated with the following formula:

$$I = w_A \, A + w_{SN} \, SN + w_{PBC} \, PBC$$

The calculation of the three basic factors in the formula is as follows:

$$A = \sum_{i=1}^{n} b_i e_i$$

$$SN = \sum_{i=1}^{n} n_i m_i$$

$$PBC \sum_{i=1}^{n} c_i p_i$$

Behavior is defined by the formula below. The effect of intention and PBC variables on the behavior variable is tested in this way.

$$B = w_I\, I + w_{PBC}\, PBC$$

b, n, c = the strength of every judgment about an outcome or quality
e, m, p = evaluation of the result and the feature
B = Behavior
I = Intention
A = Attitudes
PBC = Perceived Behavioral Control
SN = Subjective Norms
$w$ = empirically derived coefficient [75,76,78].

## 4. Results

### 4.1. Socio-Demographic and Background Variables

Table 2, below, illustrates some of the main socio-demographic and background variables of this research. Firstly, 47.3% of the people within the scope of the research are male and 52.7% are female. The vast majority (92.8%) of the people surveyed are 38 years old or younger. The vast majority (86.0%) of the households are educated at the middle or upper level. An average of 3.0 people live in the households of the people included in the study. Households generally (71.0%) have a monthly income of $1428 or less. Almost all of the households participating in the research have heard of the concept of climate change (98.8%) and are aware (96.5%).

**Table 2.** Socio-demographic and background variables of the questionnaire.

| Variables Name | Group of Variable | | Percentage ** |
|---|---|---|---|
| Gender | 1. | group: Woman | 47.3% |
| | 2. | group: Man | 52.7% |
| Age | 1. | group: 18–28 years | 52.8% |
| | 2. | group: 29–38 years | 40.0% |
| | 3. | group: 39–66 years | 7.2% |
| Descriptive statistics of age (years) Min = 18.0, Max = 66.0, Mean = 29.1, Sd = 6.9 | | | |
| Education | 1. | group: Literate (Lower than 5 years) | 4.8% |
| | 2. | group: Low (5–8 years) | 9.2% |
| | 3. | group: Middle (9–12 years) | 37.8% |
| | 4. | group: High (13+ years) | 48.2% |
| Number of households | 1. | group: 1–2 person | 23.3% |
| | 2. | group: 3–4 person | 67.6% |
| | 3. | group: 5 and + person | 9.1% |
| Descriptive statistics number of households (person) Min = 1.0, Max = 5.0, Mean = 3.0, Sd = 1.1 | | | |
| Household total income (monthly) | 1. | group: 714 $ * and less | 34.3% |
| | 2. | group: 715–1428 $ | 36.7% |
| | 3. | group: 1429+ $ | 18.5% |
| | 4. | No reply: | 10.5% |
| Have you heard of "climate change"? | 1. | group: Yes | 98.8% |
| | 2. | group: No | 0.5% |
| | 3. | group: Not sure | 0.7% |
| Are you aware of "climate change"? | 1. | group: Yes | 96.5% |
| | 2. | group: No | 0.3% |
| | 3. | group: Not sure | 3.2% |

* 1 $ = Average 7 Turkish Liras in 2020 [94]. ** All % percentage calculations are equal to 100% in the groups.

### 4.2. Results of Reliability and Confirmatory Factor Analysis

Reliability analysis was calculated using Cronbach's Alpha. All 5-Point Likert items (15) are highly reliable ($p$ = 0.000) according to the Cronbach-Alfa coefficient (0.87). CFA is a tool for accepting or rejecting measurement theory [101]. The CFA was used to test the reliability and validity of the measuring scales. Climate change-related behaviors, intentions, attitudes, subjective norms, and PBC variables are latent variables based on TPB. CFA was performed on these five main latent variables, which consist of 15 observed items. First, the goodness-of-fit index was controlled to understand the ensemble of CFAs. Table 3 shows the cutoffs for the first CFA and second (modified) CFA. Accordingly, improvements were made in the first CFA analysis and the modified CFA values were obtained. Some previous literature has accepted a value of ($\chi^2$)/df not greater than 5 [102,103]. The present study has a 5.0 value by the first CFA as seen in Equation (1).

$$\text{Chi-square } (\chi^2)/\text{df} = 471.2/94 = 5.0 \tag{1}$$

**Table 3.** The goodness of fit indices results in CFA and modified CFA.

| The Goodness of Fit Index | CFA | Modified CFA | The Goodness of Fit Criterion |
|---|---|---|---|
| CFI | 0.93 | 0.95 | $0.95 \leq$ CFI $\leq 1.00$ Perfect Fit |
| GFI | 0.87 | 0.90 | $0.90 \leq$ GFI $\leq 1.00$ Acceptable Fit |
| NFI | 0.92 | 0.94 | $0.90 \leq$ NFI $\leq 1.00$ Acceptable Fit |
| IFI | 0.93 | 0.95 | $0.95 \leq$ IFI $\leq 1.00$ Perfect Fit |
| RMSEA | 0.10 | 0.08 | $0.05 \leq$ RMSEA $\leq 0.08$ Acceptable Fit |
| RMR | 0.03 | 0.02 | RMR $\leq 0.05$ Perfect Fit |
| SRMR | 0.08 | 0.06 | $0.05 \leq$ SRMR $\leq 0.10$ Acceptable Fit |

CFI = Comparative Fit Index; GFI = Goodness of Fit Index; NFI = Normed Fit Index; IFI = Incremental Fit Index; RMSEA = Root Mean Square Error of Approximation; RMR = Root Mean Square Residual; SRMR = Standardized Root Mean Square Residual. Reference: [66].

All items (observed variables) have t-values greater than 2 as a result of the CFA. This shows that all variables are significant. However, improvements could be made to the overall functionality of the model. Check for suggested changes between variables in the CFA output file was evaluated for this reason. Modifications are suggested to address multicollinearity between variables. Although the items and their scales used in this study were reliable, the parallelism of the answers given caused the multicollinearity problem due to the closeness of the meaning of some items. Suggestions for modifying the output file after CFA are as follows: a change between A1–A2 variables would yield an improvement of 43.9 units in chi-square, 37.7 units in PBC2-PBC3, and 25.3 units in PBC1–PBC2. The improvement between the modification and 16.8 units can be seen in the improvement of B1–B2. Equation (2) shows the new value of $\chi^2$/df. Table 3 shows a comparison of goodness-of-fit indices associated with CFA and modified CFA. All of the model goodness-of-fit indicators improved after modification.

$$\text{Chi-square } (\chi^2)/\text{df} = 376.2/90 = 4.2 \tag{2}$$

Table 4 shows the item details collected from previous studies and the item's mean score on the 5-Point Likert Scale (5 = strongly agree . . . 1 = strongly disagree). The factor loadings, t-values, and $R^2$ values of all items (observed variables) used in the modified

CFA are detailed in Table 4. All items (observed variables) have t-values greater than 2 as a result of the CFA. Additional checking out scales were used for convergent validity and discriminant validity. Convergent validity turned into assessed thru the dimensions of the aspect loadings. All aspect loadings were better than 0.5 and significant ($p < 0.01$). Item convergence was turned into additionally assessed via the Average Variance Extracted (AVE) and Construct Reliability (CR). Table 4 shows that the AVE (0.5 or more suggested) and CR values (0.7 or more suggested) of the latent variables were quite sufficient for the CFA [104].

**Table 4.** Confirmatory factor analysis results (modified).

| Items (15 Items) | Mean of 5-Point Likert Scale | Factor Loads | T-Values | $R^2$ |
|---|---|---|---|---|
| **Behaviors (B)** CR = 0.895, AVE = 0.643 | | | | |
| B1—Change my driving habits to reduce my contribution to global warming and climate change. [a,b] | 3.78 | 0.60 | 9.49 | 0.23 |
| B2—I have now reduced the amount of garbage as much as possible. [a,c] | 3.07 | 0.91 | 20.62 | 0.74 |
| B3—I separate the glass/plastic/paper/battery etc. items whenever possible, for recycling. [a,c,d] | 3.51 | 0.85 | 18.41 | 0.64 |
| B4—I try to reuse objects (glass, plastic, paper, etc.). [c,b,e] | 3.22 | 0.89 | 20.40 | 0.73 |
| **Intentions (I)** CR = 0.809, AVE = 0.589 | | | | |
| I1—It is my responsibility to encourage my neighbors to notice climate change. [a] | 3.76 | 0.75 | 12.38 | 0.37 |
| I2—I am willing to adopt and apply eco-friendly practices in my daily life. [a] | 4.11 | 0.66 | 9.54 | 0.23 |
| I3—I am ready to do anything to reduce the impact of climate change. [a] | 3.63 | 0.86 | 17.15 | 0.64 |
| **Attitudes (A)** CR = 0.844, AVE = 0.643 | | | | |
| A1—Turkey's environment/nature is threatened by climate change. [a] | 4.02 | 0.80 | 11.42 | 0.35 |
| A2—Climate change negatively affects nature and wildlife in Turkey. [a] | 4.05 | 0.78 | 10.89 | 0.32 |
| A3—I am willing to pay the material and moral value for reducing climate change. [a,f] | 3.52 | 0.81 | 13.86 | 0.50 |
| **Subjective Norms (SN)** CR = 0.779, AVE = 0.542 | | | | |
| SN1—If climate change affects Turkey negatively, I would feel guilty. [a,g] | 3.73 | 0.65 | 10.61 | 0.30 |
| SN2—I feel obliged to help reduce climate change in Turkey. [a,g] | 4.01 | 0.76 | 11.99 | 0.38 |
| SN3—I think it is essential for everyone to adapt to climate change mitigation. [a] | 4.29 | 0.78 | 10.15 | 0.28 |
| **Perceived Behavioral Control (PBC)** CR = 0.849, AVE = 0.664 | | | | |
| PBC1—I believe I can contribute to mitigating the effects of climate change. [a] | 3.94 | 0.53 | 9.62 | 0.28 |
| PBC2—I can help reduce carbon emissions through the actions I take in my daily life. [a] | 3.98 | 0.91 | 11.10 | 0.27 |

Fit Indices of Path Analysis (CFA): CFI: 0.95; GFI = 0.90; NFI = 0.94; IFI = 0.95; RMSEA = 0.08; RMR = 0.02; SRMR = 0.06; ($\chi^2$)/df = 376.21/90 = 4.2. References: [a] [73] [b] [8]; [c] [79]; [d] [68]; [e] [81]; [f] [80]; [g] [81].

### 4.3. Results of Structural Equation Methods

The Structural Equation Method (SEM) was used to predict the climate change-related behavior of Turkish households under TPB. Climate change-related behaviors and intentions were endogenous latent variables, while attitudes, subjective norms, and PBC were exogenous latent variables in SEM. Attitudes, subjective norms, PBC, climate change-related behavior, and intent are expected to be in effect. Relationships are established within the framework of this model.

Therefore, in the first constructed SEM, t and normalized solutions between the variables are significant, as in the first path analysis. Likewise, the fit index of the SEM can be improved by modification. Table 5 shows the fit indices of the two models, SEM (RMSEA = 0.10, chi-square = 487.93, df = 96, $\chi^2/df$ = 5.0) and modified SEM (RMSEA = 0.08, chi-square = 357.5, df = 92, $\chi^2/df$ = 3.8).

**Table 5.** The goodness of fit results in SEM and modified SEM.

| Fit Index | SEM | Modified SEM | The Goodness of Fit |
|:---:|:---:|:---:|:---:|
| CFI | 0.93 | 0.95 | Perfect Fit |
| GFI | 0.87 | 0.90 | Acceptable Fit |
| NFI | 0.91 | 0.94 | Acceptable Fit |
| IFI | 0.93 | 0.95 | Perfect Fit |
| RMSEA | 0.10 | 0.08 | Acceptable Fit |
| RMR | 0.03 | 0.02 | Perfect Fit |
| SRMR | 0.08 | 0.06 | Acceptable Fit |

CFI = Comparative Fit Index; GFI = Goodness of Fit Index; NFI = Normed Fit Index; IFI = Incremental Fit Index; RMSEA = Root Mean Square Error of Approximation; RMR = Root Mean Square Residual; SRMR = Standardized Root Mean Square Residual. Reference: [66].

Suggested changes to improve the chi-square value in the output file are as follows: among the corrections made between the A1–A2 variables, the chi-square improved by 42.9 units, and the correction between PBC4-PBC5 improved by 41.3 units, PBC3–PBC4 improved 25.3 units and the B4–B5 improved 17.2 units. The modified values are RMSEA = 0.080, chi-square = 357.5, df = 92, $\chi^2/df$ = 3.8. This modified model is more suitable and acceptable within the fit range. The structural equation calculated from here (Equation (3)) is as follows:

$$\text{Behavior} = (1.47 \times \text{Intention}) - (0.80 \times \text{PBC}), R^2 = 0.71 \tag{3}$$

The first part of the structural equation (Equation (4)) includes the relationship between the behavioral and intention variables, which are both exogenous latent variables in the model and equations based on the assumption that the endogenous variable PBC affects behavior. Therefore, the behavior can be represented by 71% of this part of the model. Although the intent variable had a positive effect on behavior, the coefficient of the PBC variable in the SEM was negative.

$$\text{Intention} = (-0.83 \times \text{A}) + (1.74 \times \text{SN}) + (0.094 \times \text{PBC}), R^2 = 0.63 \tag{4}$$

The second part of the SEM considers the coefficients and signs of latent variables that explain intention through attitudes, subjective norms, and PBC. The variables given in the equation explained the intent variable as 63% ($R^2$ = 0.63).

Reduced from the equation (Equation (5));

$$\text{Behavior} = (-1.21 \times \text{A}) + (2.56 \times \text{SN}) - (0.94 \times \text{PBC}), R^2 = 0.24 \tag{5}$$

All these structural Equations (3)–(5) serve as visual models in Figure 3. Therefore, the climate change-related behavior of Turkish households is modeled. The insights gained from the mathematical and graphical results of the model are as follows: there was a significant positive correlation between Turkish climate change-related behavior and their

intentions (t = 6.30, standard solution = 0.77). That means Turkish households' climate change-related intentions turned into the behavior as a result of SEM. Turkish households' climate change-related intentions are affected by their subjective norm (t = 2.01, standard solution = 0.58) and PBC (t = 3.30, standard solution = 0.33). Attitudes are not turned to intentions for this model. In addition, PBC was significant on the intention while not directly significant on the behavior.

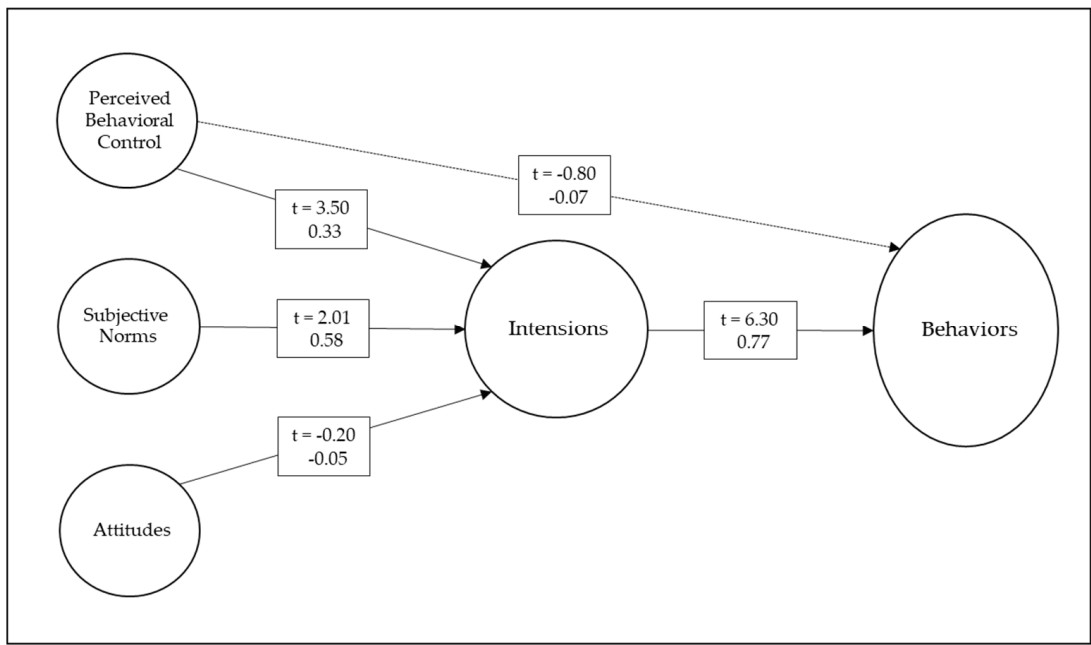

**Figure 3.** Structural Equation Model of Turkish Households' Climate Change-Related Behaviors. Fit Indices of SEM: CFI: 0.95; GFI = 0.90; NFI = 0.94; IFI = 0.95; RMSEA = 0.08; RMR = 0.02; SRMR = 0.06.

## 5. Discussion

There is a limited number of studies dealing with the behavior of households regarding climate change within the framework of TPB [8,38,39,68,72–74]. Two of them are fundamental and closer to the present study to compare the results briefly. Table 6 shows the results of the hypothesis tests of the present study, by [68,73]. Ref. [68] compared with the TPB model and the extended TPB model, with a sample of 728 Taiwan households. This research only considered the intention variable and did not include behavior in the model. The extended TPB model was found to be more effective. Accordingly, attitudes, subjective norms, and moral obligations were found to be significant in intention in the extended TPB model. On the other hand, PBC was not found to be significant in energy savings and carbon reduction behavioral intentions of Taiwanese people. RMSEA shows an acceptable fit between 0.05–0.08 and RMR shows an acceptable fit between 0.5–0.10 when both the goodness of fit of TPB and extended TPB models included in the study are examined.

**Table 6.** Comparison of the hypotheses results with other fundamental studies in the literature.

| Hypothesis | Present Study (Turkey) | [68] (Taiwan) | [73] (Malaysia) |
|---|---|---|---|
| **H1**. *The intention of the household has an impact on their climate change-related behavior.* | Accepted | - | Accepted |
| **H2**. *Attitudes towards climate change have an impact on households' intentions.* | Rejected | Accepted | Accepted |

**Table 6.** *Cont.*

| Hypothesis | Present Study (Turkey) | [68] (Taiwan) | [73] (Malaysia) |
|---|---|---|---|
| **H3**. *Subjective norms of climate change-related behavior have an impact on households' intentions.* | Accepted | Accepted | Accepted |
| **H4**. *PBC has an impact on households' Intentions.* | Accepted | Rejected | Rejected |
| **H5**. *PBC has an impact on households' Climate change-related behaviors.* | Rejected | - | - |

Ref. [73] modeled the variables that affect the pro-environmental behavior of 385 Malaysian households with TBP in the study. Accordingly, it first tested whether the intention to adapt to climate change was effective on the pro-environmental behavior of Malaysian households. In addition, the effects of attitudes towards climate change, subjective norms, and PBC variables on intention to adopt climate change were also measured. The model had an adequate fit to the data; RMSEA = 0.077, less than 0.10. In summary, household intention has an impact on climate change-related behavior in Turkey and Malaysia. Attitudes toward climate change have an impact on households' intentions in Taiwan and Malaysia but did not in the case of Turkey. Subjective norms of climate change-related behavior have an impact on households' intentions in Turkey, Taiwan, and Malaysia. PBC has an impact on households' intentions in Turkey, but not in Taiwan and Malaysia.

## 6. Strengths, Weaknesses, and Limitations of the Research

The main strength of this research is that it is the first study in Turkey to model the climate change-related behaviors of households with TPB at the national level. For this reason, it is an important guide for national and local policymakers and practitioners in the climate change field, as well as for future studies. In addition, this research also has aspects that need to be developed scientifically. Methodological improvements can be made in this research. To reach large masses in Turkey and to conduct a study at the national level, an online survey was conducted. In addition, the online survey was preferred in this study due to budget and time constraints. Online survey application makes it possible to reach more educated people as internet use also requires it. Therefore, in this study, although the population parameters generally represent the main population well, the education level was higher than expected. Therefore, it may be reasonable to plan face-to-face surveys for future studies. Again, an improvement envisaged in terms of the method should be in the form of increasing the number of questionnaires to increase the goodness of model fit and/or by using a 7 or 9 scale, instead of a 5-Point Likert Scale. Although the goodness of fit of the model within the scope of this study is within acceptable limits, it can be aimed to improve these values in future studies.

## 7. Conclusions

This research has shown that the intention of Turkish households has a positive impact on climate change-related behaviors of the households. Besides, perceived behavioral control and subjective norms have a positive impact on the intention. The climate change-related strategy for households can be developed with subjective and perceptual skills and tools to manage Turkish households' climate change-related activities.

Turkey plans to reduce its carbon emissions by 21% by 2030 within the framework of the Paris Agreement. An important pillar of carbon emission is the household. For this reason, actions for households should also be planned as a holistic precaution to be taken. This modeling research, carried out in the three provinces of Turkey with the highest population rates, shows that household intentions have a positive effect on the climate change mitigation behavior of households. On the other hand, subjective norms



and perceived behavioral control affect the intentions of households in Turkey regarding climate change. In this context, policies should be determined and implemented within the framework of these variables to reduce household carbon emissions by 2030. Within the framework of subjective norms, comprehensive public service announcements, such as "do something now, don't feel guilty in the future", which will emphasize the share of all households in climate change and invite prevention, can be effective. Again, "we can do/reduce it together" campaigns, which emphasize that measures can be taken together against climate change, can be effective in perceived behavioral control. At the end of the news bulletins, brief information about the number of carbon emissions and the effectiveness of reduction actions can be made after the weather.

As a recommendation to policymakers, it is considered important to support more comprehensive scientific projects to reduce carbon emissions from households in Turkey. In the light of the results obtained from this study, it can be suggested that future studies investigate why attitudes are not effective in climate change adopting and mitigating intentions. Again, future studies should focus on the reasons that prevent PBC from affecting behavior.

**Author Contributions:** Conceptualization, A.S.P. and Ö.C.N.; Data curation, A.S.P.; Formal analysis, A.S.P. and Ö.C.N.; Investigation, A.S.P.; Methodology, A.S.P. and Ö.C.N.; Resources, A.S.P. and Ö.C.N.; Software, A.S.P. and Ö.C.N.; Supervision, Ö.C.N.; Writing—original draft, A.S.P.; Writing—review & editing, Ö.C.N. All authors have read and agreed to the published version of the manuscript.

**Funding:** This research received no external funding.

**Institutional Review Board Statement:** All subjects gave their informed consent for inclusion before they participated in the study. In this study, the data obtained through the survey are anonymous and do not disclose any personal data. The study was conducted according to the Declaration of Helsinki. Before the survey, necessary explanations were given to the participants and their consent was obtained. Çanakkale Onsekiz Mart University Graduate Education Institute does not require ethics committee approval as obligatory according to Çanakkale Onsekiz Mart University Graduate Education and Training Regulations (Link: https://www.mevzuat.gov.tr/mevzuat?MevzuatNo=36 045&MevzuatTur=8&MevzuatTertip=5, accessed on 15 August 2022).

**Informed Consent Statement:** Informed consent was obtained from all subjects involved in the study.

**Data Availability Statement:** This research was derived from the data of the master's thesis on *Determining Consumers' Climate Change Behaviors in Turkey* in Çanakkale Onsekiz Mart University, School of Graduate Studies.

**Conflicts of Interest:** The authors declare no conflict of interest.

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
