# Peer review of "Modeling Turkish Households’ Climate Change-Related Behaviors: Theory of Planned Behavior Approach"

_sustainability, doi:10.3390/su141811290_

Round 1

Reviewer 1 Report

Dear Author(s),

Please find below my concerns and recommendations regarding your manuscript proposal entitled "Modeling Turkish Households’ Climate Change-Related Behaviors: Theory of Planned Behavior Approach".

1. I think that in the Keywords list should not appear the name of a country. Usually, the keywords refer to the research and not to a particular aspect of the article.

2. The figure 2 seems to be a little bit too big within the page. Please redraw/resize it.

3. In the Introduction chapter you should clearly define and describe the following important aspects:

- the research gap;

- the research question(s);

- the research goal.

4. After the Introduction, insert a new section entitled "Literature Review". At the end of Literature Review chapter you should present and describe the research hypotheses.

At this moment, the research hypotheses are described too late in the "2.3. Hypothesis and Analysis Methods" chapter.

5. At he rows 305-307 you say: "According to the results of first CFA; RMSEA = 0.10, SRMR=0.08, Chi-square= 471.2 and degrees of freedom (df) = 94. RMSEA values between 0.08 and 0.10 are weak and below 0.10 are completely zero. RMSEA value is acceptable when equal to or less than 0.08."

Something doesn't fit here. Your RMSEA is 0.10 and the acceptance interval is 0.08-0.10.

Please revise and correct this issue. I recommend you to read and cite in your manuscript the following reference: https://doi.org/10.3390/jtaer16040056 because here are described the cut-off values for different indicators.

6. Table 3 seems to be the same to table 5.

Please revise and correct. Do not repeat information in your manuscript proposal. Be concise.

7. Between rows 370-385 you present the coefficients of the equations. I recommend you to use a table instead of the descriptive approach.

8. In the Conclusions section please include the following important aspects:

- the managerial implications (here is the place where you can "sell" you results to the readers);

- the future research directions: based on your research findings, describe the further research directions.

Dear Author(s),

Please consider all the above remarks as being constructive recommendations in order to improve the general quality of your manuscript proposal.

Kind Regards!

Author Response

Reviewer 1 Comments and Author Notes to Reviewer

First of all thank you for devoting your precious time to improving my work. Please read the revisions and comments made in accordance with the recommendations below.

1.I think that in the Keywords list should not appear the name of a country. Usually, the keywords refer to the research and not to a particular aspect of the article.

Revision Notes: Upon the referee's suggestion, this Turkey word was removed from the keywords

2.The figure 2 seems to be a little bit too big within the page. Please redraw/resize it.

Revision Notes: Figure 1 has been resized on the recommendation of the referee.

  1. In the Introduction chapter you should clearly define and describe the following important aspects:

- the research gap;

- the research question(s);

- the research goal.

Revision notes: The correction recommended by the referee has been suggested.

  1. After the Introduction, insert a new section entitled "Literature Review". At the end of Literature Review chapter you should present and describe the research hypotheses.

At this moment, the research hypotheses are described too late in the "2.3. Hypothesis and Analysis Methods" chapter.

Revision notes: The correction recommended by the referee has been made.

  1. At he rows 305-307 you say: "According to the results of first CFA; RMSEA = 0.10, SRMR=0.08, Chi-square= 471.2 and degrees of freedom (df) = 94. RMSEA values between 0.08 and 0.10 are weak and below 0.10 are completely zero. RMSEA value is acceptable when equal to or less than 0.08."

Something doesn't fit here. Your RMSEA is 0.10 and the acceptance interval is 0.08-0.10.

Please revise and correct this issue. I recommend you to read and cite in your manuscript the following reference: https://doi.org/10.3390/jtaer16040056 because here are described the cut-off values for different indicators.

Revision notes: The correction recommended by the referee has been made. Although the reference mentioned by the referee will be quite irrelevant in the bibliography, it has been cited at the request of the referee.

  1. Table 3 seems to be the same to table 5.

Please revise and correct. Do not repeat information in your manuscript proposal. Be concise.

Revision notes: The correction recommended by the referee has been made.

  1. Between rows 370-385 you present the coefficients of the equations. I recommend you to use a table instead of the descriptive approach.

 Revision notes: There is more than one way to present a scientific data or formula to the reader. In this section, structural equation formulas that clearly reveal not only the coefficients but also the relations between the variables are given and the results are explained through the formulas. Afterwards, these relations are shown on the figure in Figure 3. Therefore, it was deemed appropriate to give this part in this way to be explanatory for future studies. Therefore, it is ensured that this part remains as it is if it is also suitable for the referee.

  1. In the Conclusions section please include the following important aspects:

- the managerial implications (here is the place where you can "sell" you results to the readers);

Revision notes: This study is a social variable-weighted study, and with the results obtained, suggestions can be made to politicians instead of companies or managers. Developed campaign and application suggestions related to the variables obtained from the study are given in the conclusion part. In addition to the referee's suggestion, another suggestion has been added to the policy makers.

- the future research directions: based on your research findings, describe the further research directions.

Revision notes: The correction recommended by the referee has been made.

  Minor spell checks are made by an authomatic grammer programme. 

Reviewer 2 Report

Dear Authors,

Thank you for the opportunity to read and review your manuscript submitted to Sustainability. After reading the manuscript, I can see that you have accomplished relevant and comprehensive research on Turkish households’ climate change-related behaviors. The research undoubtedly gives valuable findings. However, some issues stop me from being convinced that the current version is suitable for publication. Therefore, I provide a list of recommendations on how to strengthen your contribution:

1.      Please do not structurize Introduction into separate parts.

2.      Usually, figures are not provided in the Introduction. The recommendation would be to move this figure to the Appendixes.

3.      There is no need to mention basic information about Turkey (for example, the location in line 88). Your target audience supposedly has sufficient geographical knowledge.

4.      1.3 looks like a literature review. Please move this information to a separate section (2. Literature Review).

5.      Please follow the guidelines for the Introduction (provide the main am of the work, highlight the main conclusions).

6.      Please formulate a sufficient background for the formulation of the hypotheses. It is recommended to formulate hypotheses at the end of the Literature Review.

7.      It is unclear what do you mean by the sentence in line 184 („This questionnaire was also taken into account“)?

8.      There is no need to describe the chosen methods in detail (Reliability analysis, intervals of Cronbach alpha, SEM). Your reader is supposedly an academician with basic knowledge about the mentioned items. Therefore it is sufficient to mention the chosen techniques and provide a concrete rationale for their choice.

Once again, thank you for the opportunity, and I wish you good luck in strengthening the manuscript.

Author Response

Reviewer 2 Comments and Author Notes to Reviewer

First of all thank you for devoting your precious time to improve my work. Please read the revisions and

comments made in accordance with the recommendations below.

  1. Please do not structurize Introduction into separate parts.

Revision notes: The correction recommended by the referee has been made.

  1. Usually, figures are not provided in the Introduction. The recommendation would be to move this figure to the Appendixes.

Revision notes: The part and picture related to the research area have been moved from the introduction part to the material and method part.

  1. There is no need to mention basic information about Turkey (for example, the location in line 88). Your target audience supposedly has sufficient geographical knowledge.

Revision notes: The correction recommended by the referee has been made.

  1. 1.3 looks like a literature review. Please move this information to a separate section (2. Literature Review).

Revision notes: The correction recommended by the referee has been made.

  1. Please follow the guidelines for the Introduction (provide the main am of the work, highlight the main conclusions).

Revision notes: The correction recommended by the referee has been made.

  1. Please formulate a sufficient background for the formulation of the hypotheses. It is recommended to formulate hypotheses at the end of the Literature Review.

Revision notes: The correction recommended by the referee has been made.

  1. 7.      It is unclear what do you mean by the sentence in line 184 („This questionnaire was also taken into account“).

Revision notes: The correction recommended by the referee has been made.

  1. There is no need to describe the chosen methods in detail (Reliability analysis, intervals of Cronbach alpha, SEM). Your reader is supposedly an academician with basic knowledge about the mentioned items. Therefore it is sufficient to mention the chosen techniques and provide a concrete rationale for their choice.

 Revision notes:

The analyzes used in the study are normally analyzes with much longer explanations. An entry is made here for new researchers only. The information given here is briefly summarized in order to indicate the suitability of the selected methods for the research. There is no word count limit in the study.Therefore, it is ensured that this part remains as it is if it is also suitable for the referee.

Minor spell checks are made by an automatic grammer programme

Round 2

Reviewer 1 Report

Dear Author(s),

I have read the revised version of your manuscript and I appreciate your efforts to improve the quality of the article.

Now I have only one minor remark:

At section "3.2. Research Area and Sampling" you say: "A total of 405 Turkish households participated in the survey. The characteristics of the five questionnaires were unreliable or suboptimal. Thus, 400 questionnaires were found to be reliable and valid at the end of the questionnaire phase. Participants in the survey mostly (96.2%) came from the three most populous cities in Turkey, namely Istanbul (58.3%), Ankara (21.3%), and Izmir (16.6%). The three biggest provinces; Istanbul, Ankara, and Izmir, accounted for 30.5% of the Turkish population during the relevant period. There are also valid questionnaires in other cities in Turkey (3.8% of the total questionnaires). "

Please specify the period you conducted the survey, so that the readers know the time reference.

Kind Regards!

Author Response

Dear Reviewer,

First of all, thank you for your kind effort and precious labor during the article review period.

In section 3.1. there is an explanation about the survey period ''Google Forms applied a structured online questionnaire between February and July 2020. '' but if you still think we should add an additional sentence to section 3.2, then we will repeat the same information in 3.2.

Reviewer 2 Report

Dear Authors, 

Thank you for your revisions. The current version of the manuscript may be recommended for publication. 

Author Response

Dear Reviewer,

Thank you for your precious labor and kind effort to improve the article.

Best regards.